# Reproducibility study of "Robust Counterfactual Explanations on Graph Neural Networks"

## 1 Reproducibility Summary

**Scope of Reproducibility**

The aim of this paper is to reproduce the claims made in the paper *Robust Counterfactual Explanations on Graph Neural Networks* [2]. The authors claim to have developed a novel method for explaining Graph Neural Networks (GNNs) which outperforms the existing explainer methods in three different ways, by being (1) more *counterfactual*, (2) more robust to noise and (3) efficient in terms of time.

**Methodology**

The original author's code contained the code necessary to train both GNNs and explainer models from scratch. However, some alterations made by us were necessary to be able to use it. To validate the authors' claims, the trained RCExplainer model is compared with other explainer models in terms of fidelity, robustness and efficiency. We extended the work by investigating the generalisation to the image domain and verified the authors' implementation.

**Results**

For the validation of the original paper, we compare the pre-trained model and the retrained model to the results reported in the original paper. The retrained RCExplainer outperformed the other methods on fidelity and robustness, which corresponds with the results of the original authors. The measured efficiency of the method also corresponds to the original result. To extend the paper, this comparison is also performed using a train-test split, which showed no significant difference. The implementation of the metric is investigated and concerns are raised. Finally, the method generalises well to MNISTSuperpixels in terms of fidelity, but lacks in robustness.

**What was easy**

The original paper described their metrics for comparing multiple explainer models clearly, which made it easier to reproduce. Moreover, a codebase was available which included a pre-trained explainer model and files for training the other models. Because of this, we could easily find the reason for differences between our results and those of the paper.

**What was difficult**

The most difficult part of the reproduction study was determining the functionality of the provided codebase. The original authors did provide a general README file that included instructions for all code parts. However, using these provided instructions, we were not able to run this code without changes. As the provided codebase was very extensive, it was difficult to understand and determine how the different modules worked together.

**Communication with original authors**

We found it not necessary to contact the original authors for this reproduction study.

Submitted to ML Reproducibility Challenge 2021(Fall Edition). Do not distribute.

# 1 Introduction

Graph Neural Networks (GNNs) [5] are a recent development in the field of deep learning, aiming to exploit structural information by representing the input data as graphs. By passing messages along the nodes of the input graphs, these networks can use the structured nature of these graphs to reason on them. This allows GNNs to achieve groundbreaking results in a variety of fields such as the modelling of physics systems or molecular analysis [15].

However, GNNs are similar to conventional neural networks (NNs) and can therefore similarly be considered a black box. Hence, they do not always provide a sufficient *explanation* for their outcome. Nevertheless, such an explanation might be useful in some applications. An explanation, as presented in [2], is simply a subset of edges of the input graph. The authors of [2], to whom we will refer as *the original authors* from this point on, consider an explanation to be *counterfactual* if the prediction on the input graph changes significantly when the edges in the explanation are removed from the input graph.

Several methods to explain the reasoning of GNNs have already been proposed [12, 14, 10, 7]. However, these models fall short in that their generated explanations are neither *counterfactual* nor robust to noise. These features are important for a model because they make the explanations concise, easy to understand for humans and more trustworthy [2]. The original authors propose the RCExplainer model [2], which meets both criteria, and claim it is capable of outperforming existing explainer models, on the task of graph classification, while also being at least as time-efficient.

# 2 Scope of reproducibility

With this paper, we aim to validate the original authors' claims, their experimental setup, and investigate the application of their method to another domain. Our code[1] is publicly available and builds upon the code[2] of [2].

The original authors tested the RCExplainer model on three different datasets, however, due to long training times, we employed only one of these three. This reproduction paper aims to validate the following claims as made by the original authors:

- The RCExplainer model produces superior counterfactual explanations in comparison to previous methods based on fidelity scores for all levels of sparsity.
- The RCExplainer model is more robust to noise than competitive methods based on ROC AUC score.
- The RCExplainer model is at least as efficient in terms of inference time as existing explainer models.

Moreover, we conduct a set of additional experiments to inspect the following extensions to the original paper:

- Split the dataset into a proper train test split, that is no overlap between those sets, for training the explainer model and validating the effect on its performance in terms of fidelity and ROC AUC scores.
- Apply the RCExplainer method to the task of image classification using the MNISTSuperpixels dataset.
- Calculate the ROC AUC scores in two additional ways.

The next section will discuss the method of [2] in more detail and introduce our additional experiments. Section 4 reports the results to validate the original authors' claims as well as the results of our extensions. Finally, Section 5 reflects on our work and concludes that we were able to partly reproduce the original paper.

# 3 Methodology

## 3.1 Model description

The original authors propose a method consisting of two steps. First, the common decision logic of a GNN is extracted based on a set of linear decision boundaries (LDBs). This set comes from a GNN that is trained for graph classification. Second, the explainer model, based on the set of LDBs, which is a simple neural network, is trained to generate counterfactual explanations.

---

[1]Our source code is located at https://anonymous.4open.science/r/FACTAI-467E/.

[2]The original authors' code is available at https://marketplace.huaweicloud.com/markets/aihub/notebook/detail/?id=e41f63d3-e346-4891-bf6a-40e64b4a3278.

**Graph neural network** The graph neural network, denoted by $\phi$, is trained to classify input graphs. This model consists of an arbitrary number of graph convolutional layers, which produce an embedding vector, and a fully connected head. This head predicts the class probabilities from the embeddings.

**Explanation network** The explanation model, denoted by $\phi_\theta$, is trained using the embedding vectors as produced by the GNN. The network consists of two linear layers with ReLU activations.

### 3.1.1 Linear Decision Boundaries

The architecture of the classification GNN, $\phi$, can be divided into two distinct parts: the graph convolutional layers, denoted by $\phi_{gc}$, and the fully connected layers, denoted by $\phi_{fc}$. The RCExplainer model proposed by the original authors works by partitioning the output space of the graph convolutional layers into a set of decision regions, one for each class of the dataset. Given that the GNN uses piecewise linear activations on the neurons [1], its decision regions can be modelled by a set of *linear decision boundaries* (LDBs), the combination of which forms a convex polytope. As the total number of LDBs of a GNN grows exponentially with respect to the number of neurons [9], it is intractable to compute all the LDBs of a model. However, an LDB can be written as a linear equation of the form $\mathbf{w}^T\mathbf{x} + b = 0$, where the basis $\mathbf{w}$ and the bias $b$ can be computed with the following equations:

$$\mathbf{w} = \frac{\partial\left(\max_1(\phi_{fc}(\boldsymbol{\alpha})) - \max_2(\phi_{fc}(\boldsymbol{\alpha}))\right)}{\partial\boldsymbol{\alpha}}, \tag{1}$$

$$b = \max_1(\phi_{fc}(\boldsymbol{\alpha})) - \max_2(\phi_{fc}(\boldsymbol{\alpha})) - \mathbf{w}^T\boldsymbol{\alpha}, \tag{2}$$

where $\boldsymbol{\alpha} = \phi_{gc}(G)$, so the embedding of the graph $G$ in the output space of the graph convolutional layers, and the $\max_1$ and $\max_2$ operations take the highest and second-highest value of the input respectively. The original authors, therefore, propose to uniformly sample a random subset of input graphs and extract their respective LDB, in order to circumvent the complexity of computing all LDBs, giving a subset of decision boundaries $\tilde{\mathcal{H}} \subset \mathcal{H}$.

The set of LDBs forming a decision region for a specific class is then chosen to cover the maximum amount of graphs belonging to that class while ensuring that this region covers as few graphs of other classes as possible. The set of LDBs $\tilde{\mathcal{H}}_c$ that forms the decision regions of a class $c$ is determined by iteratively applying the following rule:

$$h = \min_{h \in \tilde{\mathcal{H}} \setminus \tilde{\mathcal{H}}_c} \frac{g(\tilde{\mathcal{H}}_c, c) - g(\tilde{\mathcal{H}}_c \cup \{h\}, c) + \varepsilon}{k(\tilde{\mathcal{H}}_c, c) - k(\tilde{\mathcal{H}}_c \cup \{h\})}, \tag{3}$$

where $g(\tilde{\mathcal{H}}_c, c)$ is the total number of graphs belonging to class $c$ that are covered by the LDBs in $\tilde{\mathcal{H}}_c$, $k(\tilde{\mathcal{H}}_c, c)$ is the total number of graphs *not* belonging to class $c$ that are covered by $\tilde{\mathcal{H}}_c$, and $\varepsilon$ is a small noise term that ensures the best LDB is chosen, even when the numerator equals zero. This rule is applied until $\tilde{\mathcal{H}}_c$ covers all graphs of class $c$, and then repeat this process for every class.

### 3.1.2 Explanations

Having extracted a decision region for each class, the original authors use this to generate an explanation $S$ for each graph $G$, where $S$ consists of a subset of the edges in $G$. This explanation is generated through the fully connected neural network $\phi_\theta$, parameterized by $\theta$. This model takes the node embeddings of nodes $i$ and $j$ generated by $\phi_{gc}$, and returns the probability that an edge between these two nodes is part of $G$'s explanation. Over all node pairs, this forms the matrix $\mathbf{M}$, where each entry is the probability of the corresponding edge in the adjacency matrix belonging to $S$, which is then chosen to be the set of all edges with a value greater than $0.5$ in $\mathbf{M}$.

The goal during training is to train a model such that the prediction of the GNN on the explanation is consistent with the prediction on the original graph, such that $\phi(S) = \phi(G)$. Furthermore, the original authors want to ensure that removing the edges in $S$ from $G$ changes the prediction on $G$ significantly, such that $\phi(G\setminus S) \neq \phi(G)$.

In order to satisfy these goals, the original authors define the following loss function:

$$\mathcal{L}(\theta) = \sum_{G \in D} \left(\lambda\mathcal{L}_{same}(\theta, G) + (1 - \lambda)\mathcal{L}_{opp}(\theta, G) + \beta\mathcal{R}_{sparse}(\theta, G) + \mu\mathcal{R}_{discrete}(\theta, G)\right) \tag{4}$$

where $\mathcal{L}_{same}$ is a term ensuring that the explanation of $G$ has the same classification as $G$ itself, $\mathcal{L}_{opp}$ ensures that removing $S$ from $G$ changes $G$'s classification, the combination of these terms ensuring that the explanations are counterfactual. Furthermore, $\mathcal{R}_{sparse}$ is a simple $L1$-regularization over $\mathbf{M}$, ensuring only a small amount of edges is selected to be part of the explanation by minimizing this term, and $\mathcal{R}_{discrete}$ is a term that pushed the values in $\mathbf{M}$ closer to either $0$ or $1$ to more closely resemble an actual adjacency matrix.

## 3.2   Datasets

The original paper evaluates the model on three different datasets: Mutagenicity [4], BA-2motifs [7], and NCl1 [13]. Due to time constraints, our reproducibility paper only attempts to reproduce the results on the Mutagenicity dataset. The Mutagenicity dataset is a binary dataset containing over $4000$ molecules of different sizes represented as graphs (see Table 1), with a target stating whether these molecules are mutagenic or not. Besides the Mutagenicity dataset, we also employed the MNISTSuperpixels dataset [8], containing $60,000$ graphs, in order to evaluate the RCExplainer model on a task in a different field. These graphs are obtained from the MNISTSuperpixels dataset [6], which contains images of handwritten digits, and are based on the images that are segmented using a superpixel segmentation [11]. This decreases the size of the graphs, by reducing the image from $28 \times 28$ pixels to $75$ superpixels. Furthermore, where the graph representation of a standard image would be a regular grid, where each pixel is only connected to its direct neighbours, which is identical for each image, the superpixel representation introduces irregularity between the different images, as the segmentation of each image is different ensuring each image has a different graph.

Table 1: Dataset information

| Dataset | # Samples | Avg. # Nodes | Avg. # Edges | # Labels |
|---|---|---|---|---|
| Mutagenicity | 4337 | 30 | 31 | 2 |
| MNISTSuperpixels | 60000 | 75 | 1393 | 10 |

## 3.3   Experimental setup and code

This section is split into two parts: the experiments concerning the validation of the claims made by the original paper's authors, and the experiments which validate our aforementioned extensions.

### 3.3.1   Reproducibility

First, the original authors train a GNN from scratch on the classification task. This GNN is then used to obtain the predictions and node embeddings of the input graphs. These embeddings and predictions are used to train the RCExplainer model as described in Section 3.1. Subsequently, the trained RCExplainer model is compared with other explainer models in terms of fidelity, robustness and efficiency (see Section 4.1). Due to long training times, we chose to compare the RCExplainer only to the RCExp-NoLDB [2] and PGExplainer models [7], all trained from scratch on 10 different seeds using the hyperparameters mentioned in the original paper. The GNN used as the prediction model is the pre-trained GNN provided alongside the codebase, with 3 graph convolutional layers.

Moreover, the original paper uses the entirety of the Mutagenicity dataset for training the GNN, but for training the explainer network only 1742 samples are used. We follow this same setup in our experiments. However, the original authors only mention an 80/10/10% train-val-test split for training the GNN, but no specific split for training the explanation networks. After inspecting the codebase, we observed that the training set is always a subset of the test set and, therefore, it appears that the data used for the evaluation of the RCExplainer is not entirely unseen by the model. Consequently, we decided to also evaluate all models using a train-test split of 80/20%, which is a more common split used in artificial intelligence. The results of the comparison between both splits are discussed in Section 4.2.

Furthermore, for evaluating the model based on robustness, the area under the curve (AUC) of a computed receiver operating characteristic curve (ROC) is calculated. In the provided codebase there were some unclear aspects of the AUC computation, which are addressed in Section 4.3.2.

### 3.3.2 Extension

In addition to reproducing the results of the original codebase and the original datasets, we applied the method in a different domain to evaluate the method's ability to generalise to a new domain. Where the original authors employed the Mutagenicity dataset, which requires a certain level of chemical knowledge in order to interpret the qualitative results. Therefore, we applied the RCExplainer model on the image domain as we expect these qualitative results to be easier to interpret intuitively (see 5). For this purpose, the MNISTSuperpixels dataset [8] is used. This dataset was chosen because of its relative simplicity compared to other vision datasets.

In order to apply the RCExplainer model to the MNISTSuperpixels dataset, a GNN was trained from scratch, using 4 graph convolutional layers, with 100 hidden units, followed by an embedding layer consisting of 30 units. This increase in model size is necessary to obtain results comparable to state-of-the-art [3]. More details are presented in Appendix C. We used the hyperparameters as specified in the original paper and trained the model for 600 epochs.

For comparison, both an RCExplainer and PGExplainer model have been trained to explain this GNN. The training uses the default hyperparameters for both models, similar to the comparison in the original paper. Again, following the original paper, we do not make use of a test train split, and evaluation is performed on part of the training set.

### 3.4 Computational requirements

To run all experiments, that is to say, both the reproduction study and the extension, we made use of 6 computers with varying specifications, but that contain at least one NVIDIA 2080TI GPU. The exact specifications can be found in Appendix B. Table 2 states the training time in GPU hours per model. The total training time for all models adds up to $\pm 454$ hours of GPU runtime.

Table 2: GPU computing time in hours per model. All models without superscript are trained on the Mutagenicity dataset. The $^{\dagger}$ superscript denotes models trained on the MNISTSuperpixels dataset.

| Model | RCExplainer | PGExplainer | RCExp-NoLDB | GNN$^{\dagger}$ | PGExplainer$^{\dagger}$ | **Total** |
|---|---|---|---|---|---|---|
| Time (h) | 8 | 6 | 6 | 20 | 16 | **454** |

## 4 Results

### 4.1 Results reproducing original paper

The RCExplainer is evaluated on three metrics: fidelity, robustness, and time efficiency. We compare the pre- and re-trained RCExplainer to the results reported in the original paper. For each of the metrics, the results are averaged over 10 different seeds and the standard deviations are mentioned. Note that for the pre-trained model we only have access to a single pre-trained model, so the metrics for this model are reported for only a single seed.

As mentioned in Section 3.3.1, we compare the models using two different train-test splits. In this section, we only focus on the split as the original authors did. The findings of the adjusted train-test split are discussed in Section 4.2.

**Fidelity**   The original authors use *fidelity* to compare which model produces explanations with the strongest counterfactual characteristics. Fidelity is the amount the prediction confidence decreases when the explanation is removed from the input graph. A higher value indicates stronger counterfactual characteristics. This metric can be sensitive to the sparsity of explanations, which is the percentage of the remaining edges of the input graph after deleting the explanation.

The results for this metric can be seen on the right-hand side in Figure 1. Note that the sparsity values are shown from 50% instead of 75%, because of a lack of datapoints for the PGExplainer on the 75-80% interval. Figure 1 shows that the RCExplainer has the highest performance of the models, corresponding to the findings of the original authors. However, the performance of the RCExp-NoLDB and PGExplainer in Figure 1 is significantly lower than in the original authors' paper.

As mentioned in Section 3.3.1, we use the hyperparameters as specified in the original paper. For comparison, the model was also evaluated using the hyperparameters mentioned in the README file of the codebase, changing the parameters

183  $\mu$, $\lambda$ and $\beta$ in the loss function. The corresponding results are reported in Appendix E and show that changing the
184  hyperparameters significantly affects performance. Therefore, we hypothesise that the hyperparameters are the reason
185  for the performance discrepancies as seen in Figure 1.

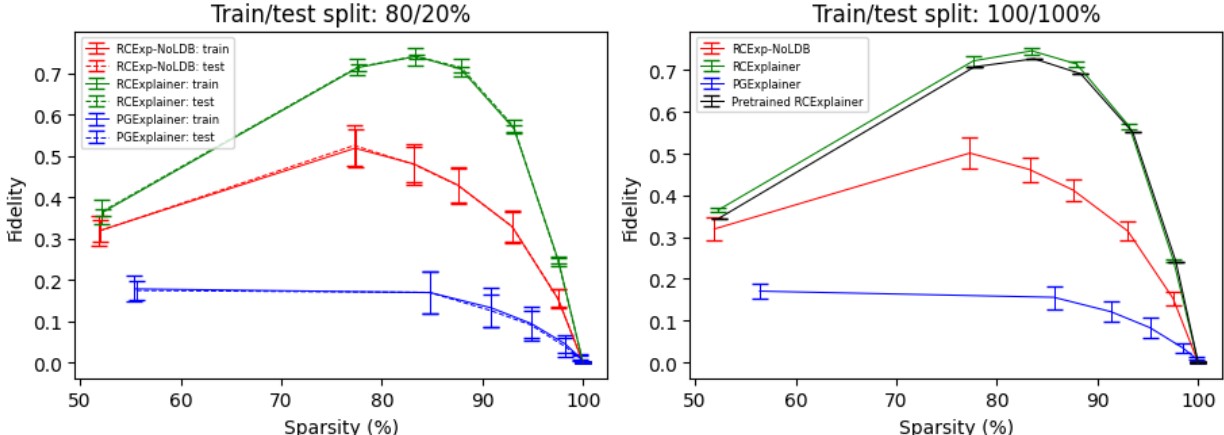

Figure 1: A comparison between different explainer models on the metric *fidelity* for two different train-set splits.

186  **Robustness**   The *robustness* of a model is measured by how much an explanation changes after noise is added to the
187  input graphs. The graphs are modified by adding random noise to the node features and randomly adding or deleting
188  edges. The produced explanation of each noisy input graph is compared to the ground truth, the $k$ best (*top-k*) edges of
189  the explanation of the unmodified graph, by computing a ROC curve and computing the AUC of this ROC curve. The
190  higher the AUC score of the model, the more robust it is.

191  Each model is evaluated for different levels of noise, measured in the percentage of nodes and edges that are modified,
192  ranging from 0% to 30%. The results are shown on the right-hand side of Figure 2. It shows that the re-trained
193  RCExplainer performs the best for almost all noise values. This corresponds with the findings in the original paper.
194  However, similar to the fidelity results, the results of the RCExp-NoLDB and PGExplainer are much lower than shown
195  in the original author's paper. We again hypothesise that this is explained by the hyperparameter tuning, following the
196  same reasoning as in the previous paragraph.

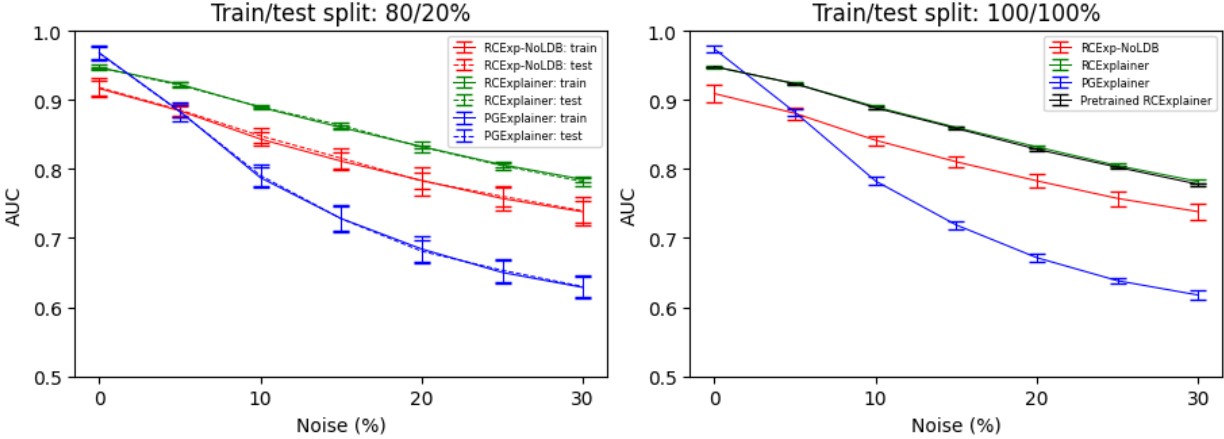

Figure 2: A comparison between different explainer models on the metric *robustness* for two different train-set splits.

197  **Efficiency**   The original authors claim their method is at least as efficient as previous methods, and report a $0.01s \pm 0.02$
198  execution time to produce a single explanation. Our experiments show a $0.007s \pm 0.0006$ execution time. This slight

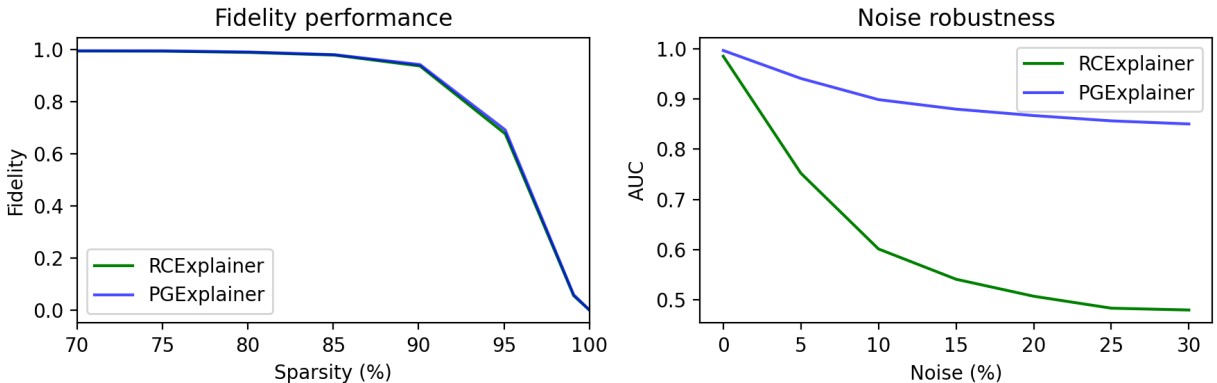

Figure 3: RCExplainer vs PGExplainer on the MNISTSuperpixels dataset. Fidelity performance shows the task is too simple, and noise robustness shows RCEexplainer is outperformed.

difference is likely due to differences in hardware platform and library versions. So, while unable to compare the performance of the RCExplainer model to other explainers, regarding their time efficiency, we were able to achieve results in line with the findings of the original authors on the run time of the RCExplainer model.

## 4.2 Results beyond original paper

As mentioned in section 3.3.1, the RCExplainer is evaluated using data that has already been encountered during training. Therefore, all models have also been evaluated on fidelity and robustness with a train-test split to see the effect of this experimental setup. Figure 1 and 2 show the results of these evaluations, where the 80/20% split is shown on the left side and the 100/100% on the right side. For both metrics, the figures show no significant differences. This lack of difference is likely because the explainer model is trained to explain the GNN, not the data, and therefore a train-test split does not seem to have a significant influence on the performance for training the explainer models.

## 4.3 Extension

This section discusses the results of our extensions to the original method. First, the results of the extension to a new domain are presented in Section 4.3.1. Then, the results of two additional AUC computations are reported in Section 3.

### 4.3.1 MNISTSuperpixels

In order to determine whether the claims of the original authors also extend to other domains, we measured the fidelity performance and noise robustness of the RCExplainer on the MNISTSuperpixels dataset (see Figure 3). To compare these curves, the same evaluation is also performed using the PGExplainer.

**Fidelity** Figure 3 show that both models achieve high fidelity, especially for sparsity lower than $90\%$, indicating that both methods saturate the task, achieving near-optimal performance.

The explainers have been trained using a 100/100% train-test split following the original paper. While this makes it significantly easier to saturate performance on the test set, as the samples are seen during training, results on other datasets in Section 4.1 show no clear difference between a more conventional train-test split and evaluating on the full set. Therefore, we hypothesise that the explainers still generalise well to this domain. Performing this evaluation with a split of 80/20% is still preferred, but not feasible in this reproduction study due to the long training time of the models.

We speculate that the decrease in fidelity for higher sparsity levels is likely not due to the model's ability to select explanations, but rather because the explanations are smaller as the sparsity level increases. As they become smaller, the counterfactual graph is more similar to the original graph retaining the same prediction. While unable to verify the performance advantage of the RCExplainer over the PGExplainer in this domain, we can verify its ability to generalise to new domains.

**Robustness**   In contrast to the fidelity performance, the noise robustness shows a clear difference according to Figure 3. This difference could be caused due to an inherent difference in robustness threshold in the MNISTSuperpixel dataset compared to Mutagenicity. As not every pixel in an image is essential, and even with large parts missing, it is still possible to correctly classify an image. The PGExplainer is more robust to noise, remaining close to the original explanation, even with noisy input graphs. However, the performance of the RCExplainer falls short, and the method appears to be less robust to noise in this domain.

### 4.3.2  AUC computation

When examining the implementation of the AUC computation we found this was adjusted when compared to the standard definition of the AUC-score, without motivation, leaving us unsure of these adjustments. The AUC-score is used to compare the accuracy of $S'$ to $S$, where $S'$ is produced from noisy input graphs to evaluate robustness to this noise. The explanation problem is formulated as a binary classification problem. For this classification, the original authors only consider *true positives* and *false positives* when measuring the AUC, discarding the *false negatives* and giving the metric a positive bias.

A false negative could occur when an edge in $S$ is no longer in $S'$, for example, when $S'$ covers a different part of the original graph. If the explainer producing $S'$ is not robust to noise, its AUC score could be incorrectly high if it only produces a subset of the ground truth explanation $S$. This means, under noisy circumstances, an explainer only has to predict a single correct edge to attain a perfect AUC score, instead of predicting the full ground truth. Therefore, false negatives appear to provide important information. *True negatives* are also discarded, but while their inclusion is standard practice, they only add information about the size of the graph compared to the explanation. When evaluating robustness, this is not as relevant and mostly reduces the difference between the scores.

Hence, we compared the original method and the inclusion of the false negatives, shown in Appendix A. For the highest noise percentage, this yields an 0.895% AUC score decrease. While this means the original method includes a slight positive bias, a bias is also present in the other explanation methods as the same evaluation code is used. Our foremost concern would be the comparison to other papers, where the metric might be implemented differently. We, therefore, chose to retain the original AUC computation method, as the bias is small and we prefer to retain the ability to compare our results to the original paper.

## 5   Discussion

This paper is a reproduction study of *Robust Counterfactual Explanations on Graph Neural Networks* [2]. We were partly able to reproduce the original authors' claims that their model produces more counterfactual explanations, is more robust to noise and is at least as time-efficient. The RCExplainer showed equal results, while the RCExp-NoLDB and PGExplainer differed, which we hypothesise is because of the hyperparameters.

For our reproduction paper, we only employed the experiments on the Mutagenicity dataset, and compared it solely to the RCExp-NoLDB and the PGExplainer, due to time constraints. Moreover, the results of the experiments have been obtained for 10 different seeds. Additionally, multiple extensions were performed to validate the experimental setup of the original paper and apply the model to the image domain.

**What was easy and what was difficult**

The original authors provided a codebase that included all code to reproduce the experiments. However, the instructions within this extensive codebase did not perfectly align with the method as proposed in the original paper. Therefore, we had to make some alterations to the code to be able to fully use it and hence mentioning all hyperparameters in the original paper would improve reproducibility. Moreover, a pre-trained explainer model was provided, but this only included a model for one seed, instead of 10 seeds. Furthermore, other explainer methods, to which the original authors compare their method were already implemented as well. Finally, the original paper described their metrics for comparing multiple explainer models clearly, which made it easier to reproduce.

**Communication with original authors**

There was no communication with the original authors, as we did not find it necessary in order to reproduce the paper.

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

## Appendix

## A AUC comparison

Table 3: AUC scores under different noise levels for RCExplainer

| Noise level | 0 | 0.05 | 0.1 | 0.15 | 0.2 | 0.25 | 0.3 |
|---|---|---|---|---|---|---|---|
| All (FN+FP+TN+TP) | 1.0000 | 0.9994 | 0.9981 | 0.9968 | 0.9960 | 0.9951 | 0.9945 |
| Original (TP+FP) | 0.9909 | 0.9512 | 0.8969 | 0.8475 | 0.8051 | 0.7622 | 0.7368 |
| False negatives (TP+FP+FN) | 0.9909 | 0.9503 | 0.8941 | 0.8429 | 0.7998 | 0.7560 | 0.7302 |
| Original and false negatives difference | 0.000% | 0.091% | 0.309% | 0.546% | 0.659% | 0.811% | 0.895% |

Table 4: AUC scores under different noise levels for PGExplainer

| Noise level | 0 | 0.05 | 0.1 | 0.15 | 0.2 | 0.25 | 0.3 |
|---|---|---|---|---|---|---|---|
| All (FN+FP+TN+TP) | 0.9996 | 0.9988 | 0.9974 | 0.9964 | 0.9945 | 0.9933 | 0.9926 |
| Original (TP+FP) | 0.9279 | 0.8810 | 0.8293 | 0.7846 | 0.7487 | 0.7179 | 0.6941 |
| False negatives (TP+FP+FN) | 0.9279 | 0.8800 | 0.8265 | 0.7809 | 0.7425 | 0.7105 | 0.6863 |
| Original and false negatives difference | 0.000% | 0.112% | 0.339% | 0.479% | 0.822% | 1.022% | 1.127% |

As concerns were raised about the specifics of the AUC computation and its effect, the AUC of different approaches are shown in Table 3 for RCExplainer and Table 4 for PGExplainer. These scores are computed on the Mutagenicity dataset using the provided pre-trained model for RCExplainer and PGExplainer, trained using the provided script and parameters. The effect is most notable under the highest noise levels, which causes $S'$ to differ the most from $S$. The original approach is positively biased for all explainers, but not equally and, therefore, affects the comparison. The effect is small enough that we chose to ignore it to retain the ability to compare to the original paper.

## B Hardware

Table 5: Hardware specifications of the machines used for training.

| | |
|---|---|
| CPU | Intel i9-9900 @ 3.10 GHz |
| GPU | NVIDIA GeForce RTX 2080 Ti |
| Memory | 64 GB |

## C MNISTSuperpixels GNN Training

For the MNISTSuperpixels dataset, we deviated from the GNN architecture used by the original authors, as it had low performance. A high accuracy of the prediction model is important because it validates the counterfactuals produced by the explanation model. A poorly trained prediction model may have arbitrary explanations, even if the explanation model is correctly trained, and therefore does not have meaningful counterfactuals. A properly trained explanation model should allow for qualitative evaluation of the method.

By increasing the number of layers and hidden dimensions of the model, the larger GNN achieves a test-set score of 85% accuracy, just short of the test-set score reached in [3]. This is shown in Figure 4. Training for the baseline model was stopped early due to low performance.

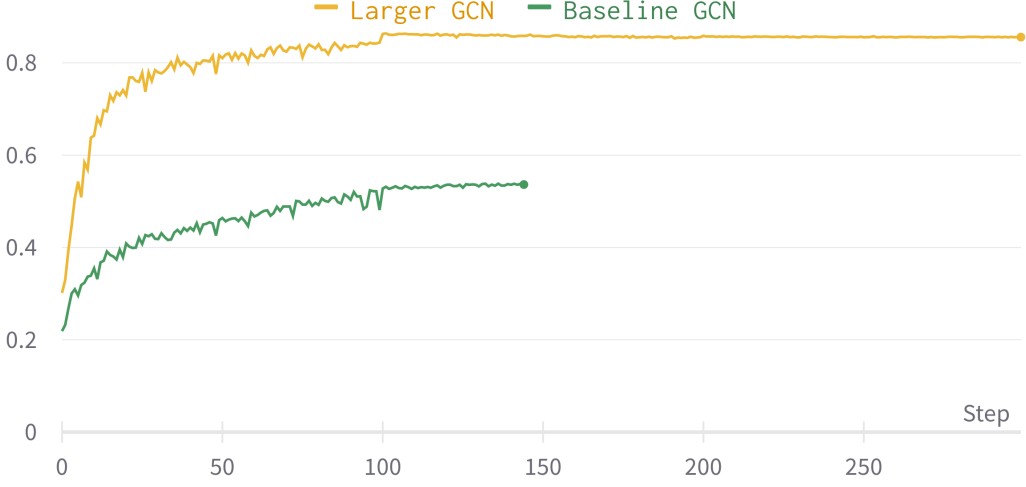

Figure 4: Validation accuracy of GNN on MNISTSuperpixels dataset

## D    MNISTSuperpixels Qualitative Results

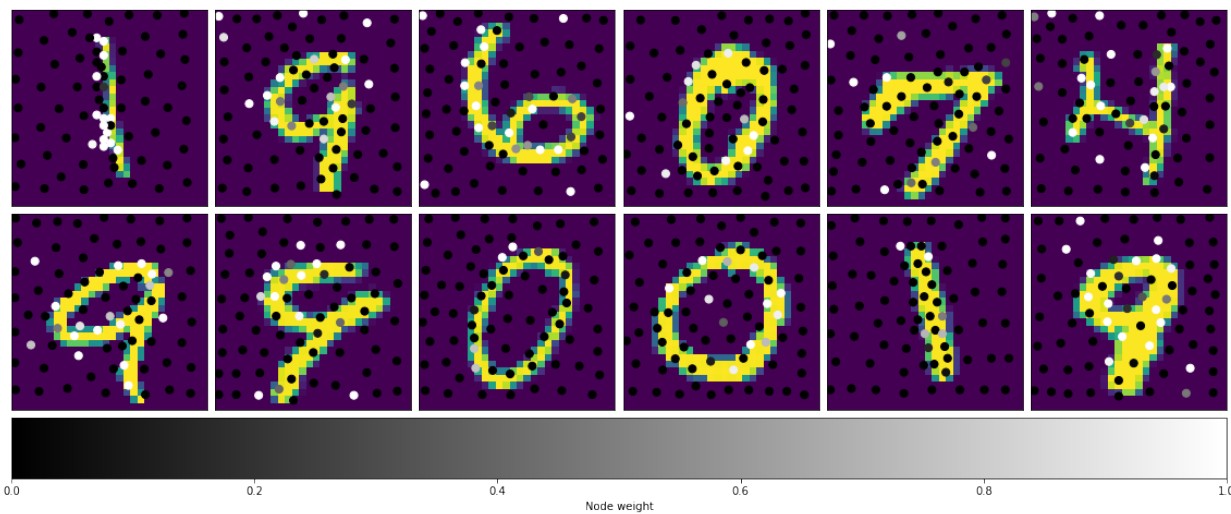

Figure 5: Node Explanations on MNISTSuperpixels dataset

Figure 5 shows the qualitative results of the RCExplainer model on the MNISTSuperpixels dataset, using twelve randomly sampled graphs. The nodes overlayed on the images are the centroids of the superpixels of the input images, and the brighter their colour, the higher their probability of being included in the explanation of the model.

While the original authors mainly define the explanation to be a set of edges they also provide a definition for an explanation consisting of nodes, which we employed for this visualization. There, a node $n \in N$ has a weight $a_n$, defined as follows:

$$a_n = \max_{i \in N}(\mathbf{M}_{ni}), \tag{5}$$

where $\mathbf{M}$ is the matrix generated by the explanation network $f_\theta$. This means that the weight of a node corresponds to the probability of the edge with the highest probability of belonging to the explanation. Every node with a weight higher than $0.5$ is then considered to be part of the explanation of that graph.

 # E Hyperparameter comparison

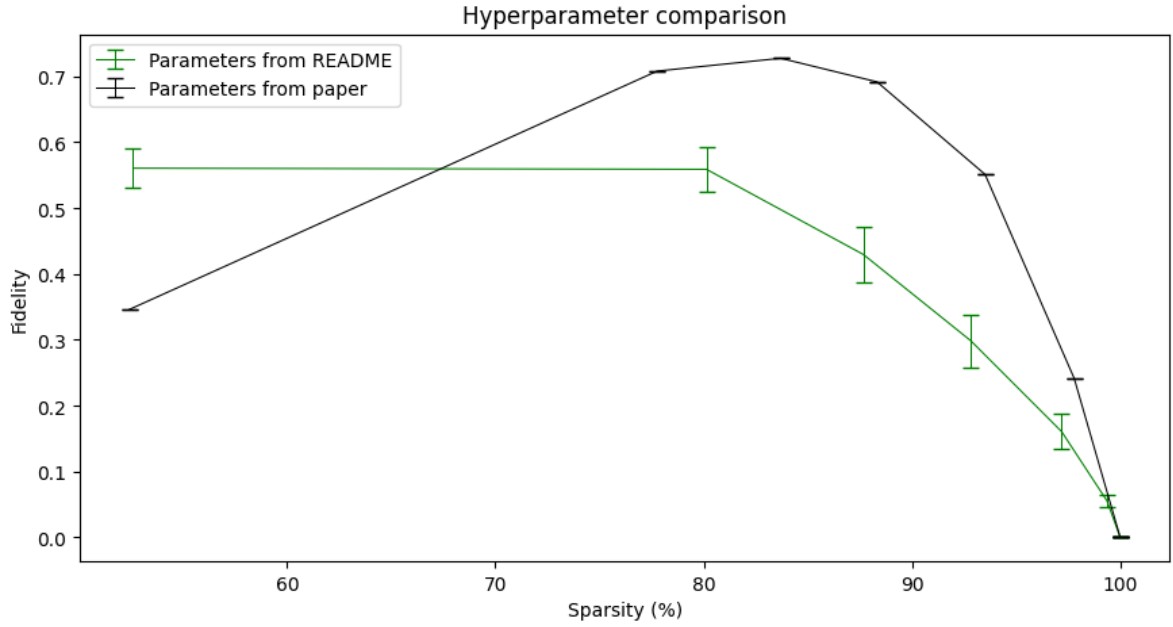

Figure 6: Comparison between two explainer models on the metric *fidelity* using a 100/100% train-test split.

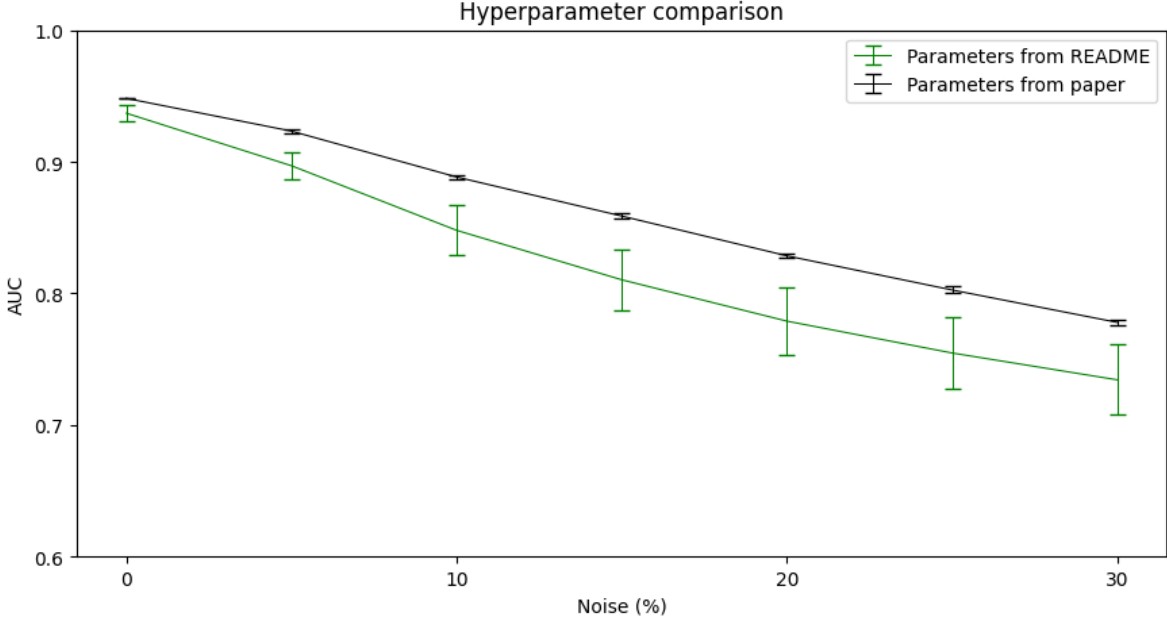

Figure 7: Comparison between two explainer models on the metric *robuustness* using a 100/100% train-test split.

