# OpenReview forum: "Reproducibility study of “Robust Counterfactual Explanations on Graph Neural Networks”"
_ML_Reproducibility_Challenge/2021/Fall — RC2021_

### Official Review · Reviewer_3oU2 · 2022-02-25
**Well written report that lacks either details on work needed on code or more in-depth analysis.**

**Rating:** 5
**Confidence:** 3

**Review:**

The reproducibility report is for the paper 'Robust Counterfactual Explanations on Graph Neural Networks', in which the experiment are replicated for the dataset Mutageneticity with a subset of the baselines. The contributed algorithm of the paper is furthermore tested on a different dataset, MNISTSuperpixels, and with slightly difference AUC measures (including false negatives).

### Scope of reproducibility
The scope is clearly stated in the introduction and is respected in the paper.

### Code
The code from original authors is re-used but modifications were required by the replicators. It is not clear what modifications were required specifically however.

### Communication with original authors
The replicators stated they did not need to contact the original authors although they struggled with the code and needed to do modifications. That is unfortunate, I am sure both authors and replicators would have benefited from it.

### Hyperparameter Search
The report only mentions effect of hyperparameters and present a simple plot with fidelity and AUC curves for 2 different sets of hyperparameters. It's unfortunate that they did not investigate further given the significance of the effect of the hyperparameters. The original code was re-used but the replicators needed to modify it to make it work.

### Ablation Study
There was no ablation study.

### Discussion on results
The report explains clearly which results were reproduced properly and which were different. They only offer hypothesis to explain why there was a discrepancy and do not attempt to study more hyperparameter values to validate it.

### Recommendations for reproducibility
The recommendations of the report are rather implicit in the sections explaining what was easy and what was difficult. Considering they had to make modifications to the code, providing clearer recommendations would have been helpful.

### Results beyond the paper
The replicators also ran the experiments on a dataset that was not used in the original paper. They also considered different versions of the AUC computation to include false negatives but found that it made only a minor difference. The explanation for the 100/100% split is not totally clear. Does it mean that the GNN was trained on all data and explanation network was subsequently trained on all data as well?

### Overall organization and clarity
The paper is well organized and easy to follow.

### Rating
Because the code was reused, I would have expected more efforts in experiments involving hyperparameter optimization or ablation study. The replicators state that they needed to modify the code but did not explain to which extend. The paper is well written and explanations are generally clear. I believe having a more in depth analysis of hyperparameter optimization or ablation study would have make it a good candidate for the journal.

---

### Official Review · Reviewer_t6QS · 2022-03-06
**Counterfactual analysis of GNNs**

**Rating:** 8
**Confidence:** 3

**Review:**

This paper seems to follow the guidelines for reproducibility, notation, code.
The authors propose a method for counterfactual explanation on GNN which uses an extraction decision boundaries shared by multiple samples. They use empirical analysis to justify their theoretical proposal.

Counterfactual analysis is important in ML generally and thus has potential for high impact in this context.  This paper is easy to follow and is well written.

The authors convincingly explore loss optimization to train a NN to produce explanations with strong counterfactual meaning.  The decision boundaries, I agree, do not seem to over fit.  I do suggest they explain more upon the extrapolation of piecewise linear neural networks to the larger family of GNNs.  I do believe this is quite novel research in the area of GNN. This paper is well-written and the it has good flow.
I would say though the noise robustness analysis should be run on at least 20 runs.

---

### Official Review · Reviewer_2AtJ · 2022-03-23

**Rating:** 8
**Confidence:** 4

**Review:**

The paper describes the reproducibility results and analysis for the "Robust Counterfactual Explanations on Graph Neural Networks" paper.

1. The paper is very well written, clear and easy to follow. The introduction clearly layed out the contributions and background of the original paper, and the scope prepares the reader for the experiments and analysis to expect. The experiments and analysis are also well written.

2. The authors conducted careful experiments in order to reproduce the results in the original paper. The authors carefully inspected hyperparamters used in thr original paper.

3. The authors also extended the experiments on the original paper on more datasets and analysis.

Overall, the paper is well written, the authors have conducted careful experiments and analysis in order to reproduce the results in the original paper. The authors even went further to extend the original paper by evaluating it on more datasets.

---

### Meta-Review · Area_Chair_C1Fs · 2022-04-07

**Recommendation:** Accept
**Confidence:** 5

**Metareview:**

The paper is well written, the reproducibility study and the extended ablations are well motivated and executed. One reviewer points out the lack of in-depth analysis in ablation studies, which could have strengthened the paper further. However, the extended analysis on a different domain dataset and special emphasis on evaluation metrics makes this paper strong, and hence I recommend to accept it.

---

### Decision · Program_Chairs · 2022-04-09

**Decision:**

Accept

**Comment:**

Following the recommendation of reviewers and meta-reviewer, the paper is accepted for ML Reproducibility Challenge 2021, and will be published in the upcoming special edition of ReScience Journal.